

# Assessment of regional and interannual variations in tropospheric ozone in chemical reanalyses

Dylan B. A. Jones[1], Lucas Prates[1], Zhen Qu[2], William Y. Y. Cheng[3], Kazuyuki Miyazaki[4], Takashi Sekiya[5], Antje Inness[6], Rajesh Kumar[3], Xiao Tang[7,8], Helen Worden[3], Gerbrand Koren[9], and Vincent Huijnen[10]

[1]Department of Physics, University of Toronto, Toronto, Ontario, Canada
[2]Department of Marine Earth and Atmospheric Sciences, North Carolina State University, Raleigh, NC 27607, USA
[3]NSF National Center for Atmospheric Research, Boulder, CO, USA
[4]Jet Propulsion Laboratory/California Institute of Technology, Pasadena, CA, USA
[5]Japan Agency for Marine-Earth Science and Technology, Yokohama 2360001, Japan
[6]European Centre for Medium-range Weather Forecasts (ECMWF), Reading, UK
[7]LAPC & ICCES, Institute of Atmospheric Physics, Chinese Academy of Sciences, Beijing, 100029, China
[8]College of Earth and Planetary Sciences, University of Chinese Academy of Sciences, Beijing, 100049, China
[9]Copernicus Institute of Sustainable Development, Utrecht University, Utrecht, the Netherlands
[10]Royal Netherlands Meteorological Institute (KNMI), De Bilt, the Netherlands

*Correspondence to*: Dylan Jones (dbj@atmosp.physics.utoronto.ca)

**Abstract.** We evaluate regional and interannual variations in tropospheric ozone in five chemical reanalyses, consisting of the Copernicus Atmosphere Monitoring Service reanalysis (CAMSRA), the second-generation Tropospheric Chemistry Reanalysis (TCR-2), the GEOS-Chem reanalysis, the Community Multiscale Air Quality (CMAQ) regional analysis, and the Chinese air quality reanalysis (CAQRA). We find that there are large regional differences (about 10–15 nmol mol$^{-1}$) in mean surface ozone between the reanalyses. GEOS-Chem has high ozone relative to the ensemble mean across most continental regions, whereas CAMSRA has low ozone. Comparison with surface ozone observations shows that the reanalyses are biased high relative to the observations, with surface ozone biases exceeding 10 nmol mol$^{-1}$ in GEOS-Chem. We find that CAMSRA has the smallest bias with respect to the observations, with negative biases in Europe, and in the central and western US, and positive biases everywhere else. In the free troposphere the reanalyses are in good agreement, and the mean bias between the reanalyses and ozonesonde observations are small, less than 4 nmol mol$^{-1}$ at 500 hPa. In addition, the correlations between the ozonesondes and the reanalyses are as high as 0.8 and 0.9 in the southern and northern midlatitudes respectively. The results suggest that chemical reanalyses should provide valuable information for quantifying variations in ozone in the free troposphere. However, to enhance the utility of the surface ozone analyses, improvements in the reanalyses are needed to better exploit assimilated observations to mitigate the impact of discrepancies in the model chemistry and ozone precursor emissions.



## 1 Introduction

Tropospheric ozone ($O_3$) plays a critical role in the chemistry of the troposphere. It is an important precursor of the hydroxyl radical (OH), the main atmospheric oxidant. It is also a greenhouse gas and an air quality pollutant. It is produced by reaction of volatile organic compounds (VOCs) and carbon monoxide (CO) in the presence of nitrogen oxides (NOx = NO + $NO_2$), and its distribution reflects the combined influence of atmospheric transport and local chemical sources and sinks. Observations of surface level ozone indicate that across eastern North America and parts of Europe surface ozone concentrations have decreased during the past two decades (Strode et al., 2015; Chang et al., 2017). Satellite observations have also revealed associated reductions in ozone precursors such as NOx and CO (e.g., Worden et al., 2013; Lamsal et al., 2015; Duncan et al., 2016; Elshorbany et al., 2024). Despite these reductions, it is estimated that 126 million people in the United States (US) in 2023 lived in counties in which ozone levels exceeded the US national ambient air quality standards (NAAQS) for ozone (EPA, 2024).

A challenge with quantifying and monitoring changes in surface ozone globally is that the surface observing network is sparse, with measurements concentrated mainly in North America, Europe, and East Asia. Space-based observations of ozone and its precursors have provided significantly greater observational coverage, starting with the Total Ozone Mapping Spectrometer (TOMS) in the 1980s (Ziemke et al., 2005) and the Global Ozone Monitoring Experiment (GOME; Burrows et al., 1999) and Measurement of Pollution In The Troposphere satellite instruments (MOPITT; Drummond 1992, Drummond et al., 2010) in the 1990s. However, satellite instruments that measure in the thermal infrared (TIR) part of the spectrum have limited sensitivity to the lower troposphere. Instruments that measure in the ultraviolet and visible (UV/VIS) or the shortwave infrared (SWIR) have greater sensitivity to the lower troposphere, but still lack sensitivity to the surface (Boersma et al., 2016), and offer limited information on the vertical distribution of trace gases in the troposphere. Satellite retrievals that combine information from the SWIR and TIR (e.g., Worden et al., 2010; Deeter et al., 2011), or the UV/VIS and TIR (e.g., Fu et al., 2013; Colombi et al., 2021, Cuesta et al., 2013 provide more information near the surface and greater constraints on the vertical distribution of trace gases. However, for these multispectral retrievals, the trace gas information near the surface reflects a blend of information from the surface and free troposphere, which complicates the use of these retrievals for interpreting variations in trace gases at the surface.

Atmospheric chemistry models provide a means of filling the observational gaps in both the surface and space-based networks, but atmospheric chemistry models typically exhibit spatially and temporally varying biases (e.g., Shindell et al., 2006; Stevenson et al. 2006; Young et al., 2013, 2018; Wild et al., 2020). Young et al. (2018), for example, found that atmospheric chemistry models typically have positive biases throughout the northern troposphere and negative biases in the southern troposphere. Parrish et al (2014) showed that models underestimate the observed long-term, decadal changes in tropospheric ozone. These discrepancies can arise from a number of issues in the models. First, the specified emission inventories in the model can have large uncertainties (e.g., Elguindi et al., 2020), which will adversely impact the fidelity of the models. Second, discrepancy in transport in the models will impact the simulated spatial distribution of ozone and its



precursors. Third, the spatial resolution of the model can also impact the model fidelity. Coarse resolution global models typically exhibit excessive stratosphere-troposphere exchange (STE), which degrades the model simulation in the upper troposphere and lower stratosphere (UTLS) (Strahan and Polansky, 2006), with implications for the lower troposphere. Lin et al. (2012) showed that in a model with a high resolution of 50 km x 50 km, episodic stratospheric intrusions could enhance

daily maximum 8-hour average (MDA8) ozone at surface sites in the western United States (US) by as much as 20–40 nmol mol$^{-1}$. They found that the intrusions enhanced the influence of the stratosphere on springtime surface ozone in the western US by a factor of 2–3 compared to previous studies using lower resolution models. Wild and Prather (2006) showed that ozone production in the planetary boundary layer (PBL) decreases with increasing model resolution. They also found that at coarse resolution, the export of ozone precursors such as NOx from continental source regions is overestimated. They

suggested that even at a resolution of 1.1° x 1.1° models will overestimate regional ozone production.

Chemical data assimilation seeks to statistically combine models with observations to obtain an improved description of the chemical state of the atmosphere. Global space-based observations are available for a suite of atmospheric trace constituents including ozone, $NO_2$, CO, nitric acid ($HNO_3$), formaldehyde (HCHO), isoprene, sulfur dioxide ($SO_2$), and aerosol optical depth (AOD). These observations, when assimilated into a model, can provide valuable constraints on

tropospheric ozone chemistry. The assimilation provides a means of correcting for discrepancies in the modeled chemical processes while the model fills the spatiotemporal gaps in the observing network. Chemical reanalyses extend this data assimilation approach in time to produce a consistent, long-term record of changes in atmospheric composition. Assimilation of satellite limb measurements for ozone profiles and nadir measurements for ozone columns has been used to evaluate ozone changes in the stratosphere and the upper troposphere (e.g., Stajner and Wargan, 2004; Jackson, 2007; Barré et al.,

2013; Emili et al., 2014). Long-term integrated data sets of stratospheric ozone have been produced by combining multiple satellite retrieval data sets (e.g., van der A et al., 2015). The first tropospheric chemistry reanalysis (TCR-1), which was conducted for 2005-2012 (Miyazaki et al., 2012, 2015), was a pioneer study for providing long-term integrated data of tropospheric composition. Global chemical reanalysis products also have been produced for 2003-2010 by the Monitoring Atmospheric Composition and Climate (MACC; Inness et al., 2013) and Copernicus Atmospheric Service (CAMS;

Flemming et al., 2017) projects. Chemical reanalyses were used to study decadal changes in NOx and CO emissions (Miyazaki et al., 2014; Jiang et al., 2017; Miyazaki et al., 2017a) and evaluate chemistry climate models (Miyazaki et al., 2017b; Kuai et al., 2020) and satellite retrievals (Cady-Pereira et al. 2017; Cuesta et al., 2018; Fu et al. 2018). The Multi-Model Multi-Component Chemistry (MOMO-Chem) framework (Miyazaki et al., 2020a) is a methodological advance that applies the same assimilation system to multiple models, and is a unique approach to provide the range of uncertainty in the

assimilation due to model errors and differences in chemistry governing air pollutant production. The recent reanalysis products have been used in various science applications (Thompson et al., 2019; Miyazaki et al., 2019; Park et al., 2020; Gaubert et al. 2020).

In support of the International Global Atmospheric Chemistry (IGAC) Tropospheric Ozone Assessment Report Phase II (TOAR-II) Chemical Reanalysis Focus Working Group, we present here an evaluation of the potential utility of the





following five chemical reanalyses, which are listed in Table 1, for quantifying regional and interannual variations in tropospheric ozone: TCR-2 (Miyazaki et al., 2020b), the Copernicus Atmosphere Monitoring Service reanalysis (CAMSRA) (Inness et al., 2019), GEOS-Chem (Qu et al., 2020), the Chinese air quality reanalysis (CAQRA) (Kong et al., 2021), and the Community Multiscale Air Quality (CMAQ) chemical reanalysis (Kumar et al, 2024). The CAMSRA, GEOS-Chem, and TCR-2 reanalyses are global, whereas CAQRA and CMAQ are regional reanalyses. A companion TOAR-II study by Sekiya et al. (2024) examines the impact of the choice of assimilated ozone and ozone precursor observations in the chemical reanalyses on the resulting ozone fields. We begin in Section 2 with a description of the five chemical reanalyses and the independent data sets used for the evaluation. The results of the evaluation are presented in Section 3, followed by a discussion in Section 4. We end with a summary in Section 5.

**Table 1.** Chemical reanalyses used in this study.

| Reanalysis system | Resolution | Assimilation scheme | Period[*] | Domain | Reference |
|---|---|---|---|---|---|
| TCR-2 | 1.1° × 1.1° | LETKF | 2005–2019 | Global | Miyazaki et al. (2020b) |
| CAMSRA | 0.75° × 0.75° | 4D-Var | 2003–2021 | Global | Inness et al. (2019) |
| GEOS-Chem | 2° × 2.5° | 4D-Var | 2006–2017 | Global | Qu et al. (2020) |
| CMAQ | 12 km × 12 km | 3D-Var | 2005–2018 | CONUS | Kumar et al. (2024) |
| CAQRA | 15 km × 15 km | LETKF | 2013–2020 | China | Kong et al. (2021) |

[*]This refers to the period for which reanalysis data was available. The work presented here focused on the 2006-2016 period, unless otherwise noted, to maximize overlap between the reanalyses.

## 2 Data and Models

### 2.1 TCR-2

The TCR-2 product employs the MIROC-Chem global chemistry transport model. The model has a T106 (1.1° x 1.1°) horizontal resolution with 32 vertical levels extending from the surface to 4 hPa. The TCR-2 meteorological fields are produced using the MIROC-AGCM (Watanabe et al., 2011) general circulation model with the model simulation nudged toward 6-hourly ERA-Interim reanalysis fields (Dee et al., 2011). The chemical mechanism in the model consists of 92 chemical species and 292 reactions. A priori anthropogenic emissions in the model are from the HTAP version 2 inventory



(Janssens-Maenhout et al., 2015). NOx emissions from soils are based on the Global Emissions Inventory Activity (GEIA) (Graedel et al., 1993). Biomass burning emissions are from the Global Fire Emissions Database version 4 (GFED4) (Randerson et al., 2018).

TCR-2 used a Local Ensemble Transform Kalman Filter (LETKF) (Hunt et al., 2007) to assimilate $NO_2$ observations from the SCanning Imaging Absorption spectroMeter for Atmospheric CartograpHY (SCIAMACHY), the Ozone Monitoring Instrument (OMI), and GOME-2; ozone observations from the Tropospheric Emission Spectrometer (TES) and the Microwave Limb Sounder (MLS); CO data from MOPITT; $HNO_3$ observations from MLS; and $SO_2$ data from OMI. The assimilated observations were used to optimize the atmospheric mixing ratio of ozone, NOx, $HNO_3$, pernitric acid ($HNO_4$),

dinitrogen pentoxide ($N_2O_5$), peroxyacetyl nitrate (PAN), and peroxymethacryloyl nitrate (MPAN). Emissions of NOx (from the surface and lightning), $SO_2$, and CO were also optimized in the assimilation. Although the model has 92 chemical species, the state vector in the assimilation consisted of a smaller subset of 35 species as well as the NOx, CO, and $SO_2$ emissions. The emissions were optimized using a state augmentation approach that used the background error covariance, determined from the forecast ensemble, to link the emissions and the atmospheric concentrations in the optimization.

**2.2 CAMSRA**

The Copernicus Atmosphere Monitoring Service reanalysis (CAMSRA) product (Inness et al., 2019) is produced using the Integrated Forecast System (IFS) of the European Centre for Medium-Range Weather Forecasts (ECMWF). The product has a horizontal resolution of T255 ($\approx 0.75°$), with 60 vertical levels from the surface to 0.1 hPa. The chemical mechanism in IFS is an extended version of the CB05 (Yarwood et al., 2005) chemical mechanism for the troposphere, which consists of

55 chemical species and 126 reactions. Explicit chemistry is not included in the stratosphere. Instead, stratospheric ozone chemistry is parameterized using the "Cariolle-scheme" (Cariolle and Déqué, 1986; Cariolle and Teyssèdre, 2007). Anthropogenic emissions in CAMSRA are from the MACCity inventory (Granier et al., 2011), with modifications to increase wintertime road traffic emissions over North America and Europe following the correction of Stein et al. (2014). Biogenic emissions are from version 2.3 of the Model of Emissions of Gases and Aerosols from Nature (MEGAN2.1;

Guenther et al., 2006, 2012) driven by meteorological fields from the Modern-Era Retrospective analysis for Research and Applications, Version 2 (MERRA-2; Gelaro et al., 2017). Soil and oceanic emissions are from the Precursors of Ozone and their Effects in the Troposphere (POET) database for 2000 (Olivier et al., 2003; Granier et al., 2005). Biomass burning emissions are from the Global Fire Assimilation System, version 1.2 (GFASv1.2; Kaiser et al., 2012).

The CAMSRA assimilation scheme is an incremental four-dimensional variational (4D-Var) data assimilation scheme

(Courtier et al., 1994) with 12-hour assimilation windows from 09:00–21:00 and 21:00–09:00 UTC. Background error covariances are diagonal so that chemical species in the control vector are optimized independently. CAMSRA optimizes ozone, CO, $NO_2$, and aerosol mass mixing ratios using observations of ozone from SCIAMACHY, the Michelson Interferometer for Passive Atmospheric Sounding (MIPAS), MLS, OMI, GOME-2, and the Solar Backscatter Ultraviolet



Radiometer (SBUV/2), together with CO observations from MOPITT and $NO_2$ observations from SCIAMACHY, OMI, and

GOME-2. Ozone precursor emissions are not optimized in the assimilation.

## 2.3 GEOS-Chem

The GEOS-Chem reanalysis (Qu et al., 2020) was produced using version v35k of the GEOS-Chem adjoint model (Henze et al., 2007). The model is driven by MERRA-2 meteorological fields at a horizontal resolution of 2° x 2.5° with 47 levels from the surface to 0.01 hPa. The model has detailed tropospheric ozone chemistry, with parameterized stratospheric

ozone based on the linearized ozone scheme of McLinden et al. (2000). A priori anthropogenic emissions in the model are from the HTAP version 2 inventory (Janssens-Maenhout et al., 2015). Biomass burning emissions are from GFED4 (Randerson et al., 2018) and NOx emissions from soils are based on Yienger and Levy (1995).

GEOS-Chem uses a 4D-Var data assimilation scheme to assimilate $NO_2$ slant column densities (SCD) from OMI to optimize NOx emissions. The GEOS-Chem reanalysis presented here assimilated OMI $NO_2$ retrievals from version 3 of the

NASA standard product OMNO2 (Krotkov et al., 2017). The assimilation minimizes a cost function that is the sum of observation-error-weighted differences between the modeled and retrieved SCDs and departures of the emission scaling factors from the prior estimates weighted by the prior emissions error. In constructing the cost function, the modeled $NO_2$ vertical column densities (VCDs) are converted to SCDs using scattering weights from the OMI retrievals. Additional details of the GEOS-Chem assimilation are discussed in Qu et al. (2020).

## 2.4 CMAQ

The CMAQ regional reanalysis (Kumar et al., 2024) uses version 5.3.2 of the CMAQ model driven by meteorological fields from version 4.1 of the Weather Research and Forecasting (WRF) model (Skamarock and Klemp, 2008). The reanalysis was conducted at a horizonal resolution of 12 km x 12 km over the contiguous United States (CONUS) with 35 vertical levels from the surface to 50 hPa. The meteorological initial and boundary conditions for WRF are from the ERA-

Interim reanalyses, while chemical initial and boundary conditions for CMAQ are from the Whole Atmosphere Community Climate Model (Marsh et al., 2013; Gettelman et al., 2019). The chemical mechanism in the model is based the Carbon Bond 6 version r3 scheme for gas-phase chemistry with the AERO7 aerosol module for aerosol processes, including secondary organic aerosols (Appel et al., 2021). A priori anthropogenic emissions in CMAQ are based on the US EPA National Emissions Inventory (NEIv2) for 2011. WRF meteorology was used with the Sparse Matrix Operator Kernel Emissions

(SMOKE) to produce meteorology-dependent anthropogenic emissions for 2011, 2014, and 2017, which were adjusted in time using EPA reported annual state-wise trends. Biomass burning emissions are from the Fire Inventory from NCAR (FINN) version 2.2 (Wiedinmyer et al., 2023). Biogenic emissions are specified using the Biogenic Emission Inventory System (BEIS).

CMAQ assimilated standard Level 2 Collection 6.1 AOD from the Moderate Resolution Imaging Spectroradiometer

(MODIS) and Version 8 of the MOPITT CO multispectral retrievals using a three-dimensional variational (3D-Var) data





assimilation scheme. The assimilation optimized total aerosol mass per mode (Aiken, accumulation, and coarse) and CO mixing ratios. Background error covariance matrices generated for January and July are used seasonally to represent wintertime and summertime background error covariances, respectively. The wintertime background error covariance matrix is used for assimilating observations between November to March, while the summertime background error covariance matrix is used during the other months.

## 2.5 CAQRA

The CAQRA product (Kong et al., 2021) is a regional reanalysis for Asia that employs the Nested Air Quality Prediction Modeling System (NAQPMS) chemical transport model (Wang et al., 2000). The model is driven by meteorological fields from WRF at a horizontal resolution of 15 km x 15 km. The meteorological initial and boundary conditions for WRF are from the NCAR-NCEP reanalysis, while chemical boundary conditions are from the Model for Ozone and Related Chemical Tracers (MOZART; Brasseur et al., 1998; Hauglustaine et al., 1998) model. The chemical mechanism in CAQRA is the carbon bond mechanism Z (Zaveri and Peters, 1999), with aqueous-phase chemistry and wet deposition based on the Regional Acid Deposition Model (RADM) mechanism from version 4.6 of CMAQ and inorganic aerosol processes represented by ISORROPIA1.7 (Nenes et al., 1998). Anthropogenic emissions in the model are from the HTAP version 2.2 inventory with a 2010 base year (Janssens-Maenhout et al., 2015). Emissions of VOCs are from the MEGAN-MACC model (Sindelarova et al., 2014). Biomass burning emissions are from GFED4 (Randerson et al., 2018), soil NOx emissions are from the Regional Emission Inventory in Asia (Yan et al., 2003), and oceanic emissions are from the POET database (Granier et al., 2005).

CAQRA uses an LETFK data assimilation scheme to assimilate surface observations of ozone, CO, $NO_2$, $SO_2$, $PM_{2.5}$, and $PM_{10}$ to optimize the atmospheric concentration of these constituents. In constructing the background error covariance, inter-species correlation is neglected. Thus, in the assimilation, each chemical species is optimized using only observations of that species. The assimilation also employs species-specific inflation factors that vary in space and time. Additional details of the CAQRA assimilation configuration are available in Kong et al. (2021).

## 2.6 Ozonesondes

To evaluate the reanalyses, we use ozonesonde observations from the TOAR-II Harmonization and Evaluation of Ground-based Instruments for Free Tropospheric Ozone Measurements (HEGIFTOM) effort (https://hegiftom.meteo.be/datasets/ozonesondes). The measurement precision for the ozonesondes is better than 3–5% with an accuracy of about 5–10% (Smit et al., 2007). The HEGIFTOM ozonesonde data were harmonized to remove systematic biases and to provide an uncertainty estimate for every measurement. The database contains time series observations from 43 sites. Here we use data from 39 sites that each had a minimum of 48 measurements between January 2003 and December 2022.



### 2.7 Surface ozone observations

Surface ozone observations from the TOAR-I data set (Schultz et al. 2017) were used for evaluation of the reanalyses. The dataset consists of a consistent long-term record (1990–2014) of surface ozone observations that have been harmonized and processed with consistent quality control. We used the 2° x 2° gridded monthly mean observations for the period 2003-2014 for comparison with the reanalysis. All of the reanalyses were regrided onto the ozone 2° x 2° grid for the evaluation. The ozone data are available for both rural and urban sites. However, since the lowest resolution of the available reanalyses is 2° x 2.5°, which cannot reliably distinguish between urban and rural locations, we follow the approach of Huijnen et al. (2020) and Sekiya et al. (2024) and use only rural TOAR-I observations in the evaluation of the reanalyses.

### 3 Results

#### 3.1 Climatological ozone distribution

The mean ozone distribution at the surface from the five reanalyses for 2006-2016 and the differences between the individual reanalyses and the ensemble mean are shown in Fig. 1. The global mean distribution (Fig. 1a) consists of a band of high ozone across the northern subtropics and a minimum in the tropics. In North America, there are high values over the southeastern US and the mountain west. In Asia, high ozone values are located over northern India. Figure 1b shows that GEOS-Chem is high everywhere relative to the ensemble mean, with differences exceeding 10 nmol mol$^{-1}$ in Asia and western North America. In contrast, CAMSRA is low over all continental regions, with the largest difference of about 10-15 nmol mol$^{-1}$ over central Africa and parts of Asia. TCR-2 is slightly lower than the ensemble mean at the high latitudes and about 5–10 nmol mol$^{-1}$ higher over tropical South America and central Africa. For the regional reanalyses, CAQRA is about 5 to 15 nmol mol$^{-1}$ lower across much of Asia, with small positive differences over Thailand and Myanmar, whereas CMAQ had small differences of less than 5 nmol mol$^{-1}$ across much of the US, with positive differences in the Pacific Northwest and negative differences in the southeastern US. Examination of the seasonal differences between the individual reanalyses and the ensemble mean (Fig. S1) reveals that the pattern of differences shown in Fig. 2 is relatively consistent seasonally in all reanalyses, with the exception of CMAQ, although the magnitude of the differences is generally larger in the winter hemisphere. CMAQ has positive differences across much of the US in December–February (DJF), and negative differences in June–August (JJA).

In the middle troposphere, the mean ozone distribution, shown in Fig 2a, is similar to that at the surface, with a band of high ozone across the northern subtropics, but with fewer small-scale features than at the surface. Note that only the global reanalyses are evaluated in the free troposphere since only surface fields are available from the regional reanalyses. Examination of the differences between the individual global reanalyses and the ensemble mean (in Fig. 3a) shows that the reanalyses are all closer to the mean in the middle troposphere. The mean differences in the individual reanalyses are generally less than 5 nmol mol$^{-1}$. For all three global reanalyses, the mean differences with respect to ensemble mean are



mainly in the tropics and subtropics, with TCR-2 and GEOS-Chem exhibiting an opposite pattern of differences. TCR-2 has a large difference with respect to the ensemble mean over the tropical Atlantic, northern South America, and the Indian Ocean, whereas GEOS-Chem is slightly different over the tropical Atlantic, northern South America, and the Indian Ocean. CAMSRA has small positive differences over the Pacific and small negative differences over Africa and the southern tropical Atlantic and Indian Oceans. The pattern of the differences in the reanalyses is consistent seasonally (see Fig. S2) but with a seasonally-dependent latitudinal shift. In GEOS-Chem and TCR-2 the difference pattern is shifted into the tropics of the summer hemisphere in a manner similar to the seasonal shift of the Intertropical Convergence Zone (ITCZ).

At 250 hPa the mean ozone distribution shown in Fig. 2b reveals the influence of the extratropical lower stratosphere with high ozone at the high-latitudes. In the tropics there is a minimum in ozone over the warm pool of the tropical western Pacific, which reflects the influence of convective transport (e.g., Pan et al., 2017). The mean differences between the individual reanalyses and the ensemble mean (Fig 3b) are relatively small, with the largest differences mainly in the southern midlatitudes. The spatial pattern of the differences in TCR-2 is opposite that in GEOS-Chem, which has high ozone in the southern midlatitudes relative to the ensemble mean, while TCR-2 is low in the southern midlatitudes. The high ozone in GEOS-Chem extends across the southern extratropics in DJF and MAM (see Fig S3), whereas it is confined to the southern midlatitudes in JJA and SON. In TCR-2, the differences in the southern midlatitudes are largest in JJA and SON. CAMSRA has high ozone relative to the ensemble mean in the northern high-latitudes in JJA and low ozone in the southern high-latitudes in DJF.

MDA8 ozone is a daily metric widely used for air quality standards and for ozone exposure studies (e.g., Turner et al., 2015; Flemming et al., 2018; Lyu et al., 2019; Chen et al., 2024). The ensemble mean MDA8 ozone distribution and the differences between the individual reanalyses and the ensemble mean are shown in Fig. 4. The spatial pattern of differences in MDA8 (Fig. 4b) is similar to that shown for ozone in Fig. 1b, except for the CMAQ reanalysis. In CMAQ, MDA8 ozone is high relative to the ensemble mean over the western and central US, in contrast to the pattern of differences in mean ozone shown in Fig. 1b. Overall, we find that the regional differences in MDA8 ozone between the individual reanalyses and the ensemble mean are generally smaller than the differences shown for surface ozone in Fig. 1b. The exception is the TCR-2 reanalysis which has larger MDA8 differences with respect to the ensemble mean over South America and central Africa.

## 3.2 Regional ozone variations

The seasonal variations in regional mean ozone are shown in Fig. 5 for the nine regions defined in Table 2 (and shown in Fig S4). At the surface (Fig 5a), the seasonal cycle is consistent across all the reanalyses. In all regions, CAMSRA has the lowest ozone mixing ratios. For example, in the US, ozone in CAMSRA is about 5 nmol mol$^{-1}$ lower than in TCR-2. In contrast, GEOS-Chem ozone is 5–10 nmol mol$^{-1}$ higher than in TCR-2. Indeed, GEOS-Chem has the highest ozone concentrations in all regions except for South America and the US. In South America, TCR-2 has higher mean ozone than



GEOS-Chem in all months except for July and August. In the US, CMAQ is higher than GEOS-Chem between December-March, whereas in China, CAQRA is fairly consistent with CAMSRA.

In the free troposphere, at 500 hPa (Fig 5b), the seasonality of the global reanalyses is generally consistent in the extratropics, which is in agreement with the results shown in Fig. 3a. In the tropics, there are larger discrepancies between the reanalyses over South America, North Africa, and the Middle East. Over South America, for example, TCR-2 has higher

ozone concentrations between January–June and September–December, similar to the discrepancies observed at the surface (Fig 5a). Over the US, ozone concentrations peak about three months later in GEOS-Chem than in the other reanalyses. And over China, GEOS-Chem has a broader ozone maximum than CAMSRA and TCR-2. In the UTLS (Fig. 5c), TCR-2 and CAMSRA are generally in agreement in terms of the ozone concentrations and variability in all regions, probably due to the assimilation of ozone measurements from MLS (Sekiya et al., 2024). However, over the US and Europe, ozone in GEOS-

Chem is low relative to TCR-2 and CAMSRA. In addition, over South America and southern Africa, GEOS-Chem exhibits a larger seasonal cycle than CAMSRA and TCR-2. The discrepancy could be attributed to the lack of assimilation of ozone measurements in GEOS-Chem.

The time series of the regional mean ozone concentrations are shown in Fig. 6. As can be seen in Fig 6a, the lower ozone mixing ratios at the surface in CAMSRA are present in all years. The differences in the mean concentrations between

CAMSRA and the other reanalyses is particularly pronounced in the three African regions. Across most regions, with the exception of South America, the three global reanalyses have similar interannual variability in surface ozone, with a high ozone bias in GEOS-Chem and a low ozone bias in CAMSRA. In South America, ozone in TCR-2 is generally higher than 20 ppb throughout the year, while CAMSRA ozone is lower than 10 ppb during the summer months. For the regional reanalyses, we find that in the US, surface ozone variability in CMAQ is similar to that in GEOS-Chem, whereas in China,

CAQRA is similar to CAMSRA. In the northern extratropical middle troposphere, over the US, Europe, and China (Fig 6b), the variability in the three global reanalyses is remarkably similar. The worst agreement is found over Northern Africa, where the interannual variability is significantly different in each of the three global reanalyses. Over South America and central Africa the simulation of the ozone maxima in TCR-2 is consistent with the other reanalyses, but TCR-2 significantly overestimates the ozone minima. In the UTLS (Fig. 6c), Northern Africa is the region with the greatest disagreement

between the reanalyses in their simulation of the ozone variability. The reanalyses are in good agreement in all other regions. In general, the simulated variability is most different in GEOS-Chem. For example, over Europe and the US, GEOS-Chem underestimates the ozone maxima (also seen in Fig. 5c) and fails to reproduce the year-to-year variability in the ozone maxima simulated by CAMSRA and TCR-2.

The linear trend in surface ozone in the reanalyses is shown in Fig. 7. There are large regional differences between the

reanalyses in the trends. For example, in Europe, GEOS-Chem has positive trends of 0.1–0.2 nmol mol$^{-1}$ per year everywhere, whereas TCR-2 has negative trends that are comparable in magnitude. In CAMSRA there are negative trends in southern and eastern Europe and positive trends in parts of northern Europe. In Asia TCR-2 has positive trends (that are as much as 0.4 nmol mol$^{-1}$ per year), whereas CAMSRA has large negative trends in East Asia (exceeding −0.4 nmol mol$^{-1}$ per



year) with small positive trends in South Asia. In North America, the pattern of the trends is similar in CAMSRA, GEOS-
Chem, and CMAQ, with negative trends over the southeastern US and the mountain west, and positive trends over northern
and eastern Canada. However, in TCR-2 there are negative trends over much of North America.

**Table 2.** Regional definitions used in the ozone evaluation.

| Region | Latitude range | Longitude range |
|---|---|---|
| United States | 28.0°N–50.0°N | 70.0°W–125.0°W |
| Europe | 35.0°N–60.0°N | 10.0°W–30.0°E |
| India | 8.0°N–33.0°N | 68.0°E–89.0°E |
| China | 30.0°N–40.0°N | 110.0°E–123.0°E |
| Middle East | 12.5°N–37.5°N | 30.0°E–60.0°E |
| Northern Africa | Equator–20.0°N | 20.0°W–40.0°E |
| Central Africa | Equator–20.0°S | 10.0°E–40.0°E |
| Southern Africa | 22.0°S–31.0°S | 25.0°E–34.0°E |
| South America | Equator–20.0°S | 50.0°W–70.0°W |

## 3.3 Evaluation with independent observations

The mean surface ozone observations from TOAR-I for 2006-2014 and the differences between the individual
reanalyses and the TOAR observations are plotted in Fig. 8. The evaluation only extends to 2014 because observations
TOAR-I database are not available after 2014. In addition, the CAQRA reanalysis was not included in this evaluation
because of the short temporal overlap between CAQRA and the TOAR-I database and the limited number of TOAR-1
observations over Asia. As can be seen in Fig 8b, GEOS-Chem, TCR-2, and CMAQ are biased high relative to the TOAR
observations, with GEOS-Chem exhibiting the highest global bias of about 14 nmol mol$^{-1}$. The global mean bias for
CAMSRA and TCR-2 is 1.6 nmol mol$^{-1}$ and 5.5 nmol mol$^{-1}$, respectively. Over the US, CMAQ has a mean high bias of 6.4
nmol mol$^{-1}$. The CAMSRA global mean bias reflects the compensating influence of negative biases over the central and
western US and over central and eastern Europe, and positive biases everywhere else. Examination of the seasonality of the
bias (see Fig. S5) reveals that in GEOS-Chem the largest global mean bias with respect to the TOAR observations is 16.6
nmol mol$^{-1}$ in JJA and the smallest mean bias is 10.5 nmol mol$^{-1}$ in DJF. In TCR-2 and CAMSRA the largest mean bias is



9.8 nmol mol$^{-1}$ and 5.3 nmol mol$^{-1}$, respectively, in JJA and the smallest mean bias is 1.7 nmol mol$^{-1}$ and −1.6 nmol mol$^{-1}$, respectively, in DJF. In CMAQ the largest total mean bias is 10.4 nmol mol$^{-1}$ in DJF, while the smallest total mean bias is 4.3 nmol mol$^{-1}$ in JJA.

335 A comparison of the global reanalyses with ozonesonde data in the middle troposphere and UTLS is shown in Fig. 9. The ozonesonde data were binned into the following three latitude bins: 20°S–60°S, 20°S–20°N, and 20°N–60°N. At 500 hPa the reanalyses capture the variability in the ozonesonde data well, with correlations of about 0.8 and 0.9 in the southern and northern midlatitudes, respectively. The correlation is lower in the tropics, with values between 0.5–0.7. The time series of the reanalyses at the individual ozonesonde sites are shown in Figs. S6 and S7. The higher correlation in the midlatitudes

is expected since ozone transport in this region is dominated by large-scale synoptic processes that are well constrained by meteorological reanalyses. In contrast, transport in the tropics is dominated by convective processes, which are less well constrained by models and meteorological reanalyses. The mean bias across all three regions in the middle troposphere is less than 4 nmol mol$^{-1}$. In the northern midlatitudes, the largest bias, which is in the GEOS-Chem reanalysis, is only about 2 nmol mol$^{-1}$. The standard deviation is also low, less than 4 nmol mol$^{-1}$, in the tropics and northern midlatitudes. The standard

deviation increases to about 6 nmol mol$^{-1}$ in the southern midlatitudes. However, it should be noted that there are only three ozonesonde sites in the southern midlatitude region.

 In the UTLS (at 250 hPa) the correlations in the midlatitudes are just as high as at 500 hPa. However, the correlations in the upper tropical troposphere are slightly lower. In the northern midlatitudes, the largest mean bias is about 10 nmol mol$^{-1}$, which is relatively small given that the mean ozone mixing ratio varies from about 75 nmol mol$^{-1}$ at 20°N to about 200 nmol

mol$^{-1}$ near 60°N (see Fig. 2b). In the tropics, where the ozone mixing ratio is lower, the mean bias is smaller. The largest mean bias, exceeding 20 nmol mol$^{-1}$, is found in GEOS-Chem in the southern midlatitudes. The standard deviation in the UTLS is larger than at 500 hPa, particularly in the midlatitudes, which is expected given the higher ozone mixing ratios in the lower stratosphere and the small-scale meteorological processes that drive variability in the tropopause region. At 250 hPa the ozone distribution will be strongly influenced by variations in the tropopause and it is a challenge for models to

reproduce the small-scale, dynamically-driven variations in ozone in this region.

## 3.4. Surface NO₂ distribution

 As NO$_2$ is a key ozone precursor, examination of the NO$_2$ distribution in the reanalyses could provide insight in the source of the differences in surface ozone between the reanalyses. The ensemble mean NO$_2$ distribution and the differences between the individual reanalyses and the ensemble mean are plotted in Fig. 10. As shown in Fig. 10b, the ensemble mean

reflects the combined influence of low NO$_2$ mixing ratios in GEOS-Chem and high NO$_2$ in CAMSRA. CMAQ has low NO$_2$ in the US, whereas CAQRA generally has high NO$_2$ in eastern China and low NO$_2$ in western China, northern India, and southeast Asia. The seasonal variations in regional mean NO$_2$ at the surface are shown in Fig. 11. Across all region, GEOS-Chem has low NO$_2$ seasonally compared to the other reanalyses. For most regions, except South America and China, CAMSRA has higher NO$_2$ than TCR-2. For example, over North America, India, and the Middle East, CAMSRA has much





higher NO$_2$ than TCR-2. The large discrepancies in surface NO$_2$ between the reanalyses is due, in part, to the differences in the assimilation configuration employed in the reanalyses. For example, GEOS-Chem constrained only NOx emissions, CAMSRA constrained NO$_2$ and ozone mixing ratios, TCR-2 constrained NO$_2$ and ozone mixing ratios as well NOx emissions, and CMAQ constrained aerosol mass and CO mixing ratios. In addition, GEOS-Chem has the lowest spatial resolution at 2° x 2.5°. These differences will have a significant impact on the simulated surface NO$_2$ in the reanalyses since

NOx emissions represent a strong forcing on surface NO$_2$ mixing ratios in the assimilation, with implications for surface ozone.

## 4. Discussion

The larger biases in ozone at the surface compared to the free troposphere is consistent with previous studies that showed that atmospheric chemistry models tend to overestimate surface ozone concentrations (e.g., Reidmiller et al., 2009;

Travis et al., 2016). The large positive mean biases of up 10–15 nmol mol$^{-1}$ found here are similar to those reported by Young et al (2018) in their evaluation of models used in the Atmospheric Chemistry and Climate Model Intercomparison Project (ACCMIP). One potential contributor to the overestimate of surface ozone in the reanalyses could be the model resolution, since according to Wild and Prather (2006), even at a horizontal resolution of 1.1° x 1.1° models will overestimate regional ozone production. Furthermore, it was suggested by Valin et al. (2011) that high spatial resolution of

4–12 km is required to capture the non-linear NOx chemistry. CMAQ was run at a high spatial resolution of 12 km x 12 km, however only aerosol optical depth and the CO mixing ratio were optimized in the analysis. Thus, in CMAQ, ozone was adjusted indirectly through changes in the model chemistry. TCR-2 and CAMSRA both optimized the ozone mixing ratio, but the spatial resolution of both models, 1.1° and 0.75°, respectively, is still coarse. Higher resolution is also important to more effectively assimilate the satellite observations to analyze emissions and concentrations on a megacity scale (Sekiya et

al., 2021). GEOS-Chem has the lowest spatial resolution (at 2° x 2.5°) and the highest ozone bias. Another possible source of the high surface ozone in GEOS-Chem is bias in the chemical mechanism in the model. Qu et al. (2020) found that using the a posteriori NOx emissions from the GEOS-Chem reanalysis in a more recent version of the model resulted in lower ozone mixing ratios, which they attributed to differences in the chemical mechanism and VOC emissions in the two versions of GEOS-Chem.

The regional discrepancies in surface ozone between the reanalyses clearly reflects the fact that atmospheric composition measurements from space have less sensitivity to ozone and its precursors near the surface. However, the discrepancies in surface ozone in the reanalyses also reflect differences in the assimilation configuration employed in the reanalyses as well as discrepancies in the chemical mechanisms used in the models. Near the surface the ozone lifetime is shorter than in the middle and upper troposphere, thus, near-surface information from the observations ingested in the

assimilation will be rapidly destroyed and the surface ozone analysis will be strongly influenced by discrepancies in the model chemistry and precursor emissions at the surface. As a result, the configuration of the assimilation will have a





significant impact on the surface ozone analysis. For the reanalyses considered here, GEOS-Chem optimized only NOx emissions, CAMSRA optimized the atmospheric concentration of ozone and its precursors, CMAQ optimized the atmospheric concentration of aerosols and CO, and CAQRA assimilated only in situ surface observations, but optimized the

atmospheric concentration ozone, CO, $NO_2$, $SO_2$, and particulate matter. Zhang et al. (2019) showed that optimizing only NOx emissions can result in large regional differences in the inferred emissions compared jointly optimizing the emissions together with the ozone concentrations due to the influence of discrepancies in the ozone field on the $NO_2$ concentrations. TCR-2 is the only reanalysis that optimized both the atmospheric concentrations of ozone and its precursor as well as the precursor emissions. To mitigate the impact of discrepancies in model chemistry and ozone precursor emissions,

improvements in the chemical reanalyses are required to jointly optimize the atmospheric concentrations of ozone and its precursors together with the ozone precursor emissions. In this context, the construction of better background error covariances in the reanalyses is needed to propagate the information from the observations in space, time, and, in particular, across species.

In our evaluation of the reanalyses, we have focused on discrepancies in $NO_2$ as possible factor influences the ozone
analysis. However, VOCs also play an important role in ozone formation and none of the reanalyses examined here assimilated observations to constrain the VOC emissions. Isoprene is the dominant non-methane VOC (Guenther et al., 2012) and HCHO is a key byproduct of isoprene oxidation. Assimilation of satellite observations of HCHO have been widely used to quantify isoprene emissions (e.g., Palmer et al., 2003; Millet et al., 2008; Stavrakou et al., 2015; Kaiser et al., 2018). In addition, satellite observations of isoprene are now available from the Cross-Track Infrared Sounder (CrIS)
satellite instrument and can provide constraints on isoprene emissions (Wells et al., 2020). Integrating HCHO and isoprene observations should provide greater constrains on tropospheric ozone. For the first time, observations of ozone, $NO_2$, and HCHO are also available from geostationary (GEO) orbit by the Geostationary Environment Monitoring Spectrometer (GEMS) and the Tropospheric Emissions: Monitoring of Pollution (TEMPO) instruments. In contrast to low-earth orbiting (LEO) satellites, GEO instruments greater daytime temporal coverage to capture diurnal variations in ozone, $NO_2$, and
HCHO. Park et al. (2024) and Hsu et al. (2024) showed that assimilating GEO observations of $NO_2$ results in improved NOx emission estimates compared to those inferred from assimilating LEO data. Integrating GEO and LEO observations will enhance the constraints on tropospheric ozone in the reanalyses. Ultimately, given the limitations in the vertical sensitivity of the satellite measurements near the surface, and the limitations in the observational coverage of the surface network, integrating the surface and satellite observations in the reanalyses to exploit their complementarity will provide valuable
constraints on surface ozone variability.

## 5. Summary

We have conducted an evaluation of the regional and interannual variations in tropospheric ozone in five chemical reanalyses, consisting of three global (TCR-2, CAMSRA, and GEOS-Chem) and two regional (CMAQ and CAQRA)



reanalyses. We found that at the surface there can be large regional differences in mean ozone (exceeding 10 nmol mol$^{-1}$)
between the reanalyses. In general, the GEOS-Chem reanalysis was biased high relative to the ensemble mean across most
continental regions, whereas CAMSRA was biased low. The TCR-2 reanalysis was closest to the ensemble mean at the
surface except in tropical South America and central Africa. In the free troposphere the global reanalyses were in closer
agreement. At 500 hPa the mean bias between individual reanalyses and the ensemble mean was less than 5 nmol mol$^{-1}$ in
most regions. Similarly, in the UTLS the mean bias between the individual reanalyses and the ensemble mean was less than
5 nmol mol$^{-1}$, except in the northern high-latitudes where the biases exceeded 30 nmol mol$^{-1}$, with GEOS-Chem and
CAMSRA exhibiting large positive bias and negative biases, respectively.

Regionally, at the surface the reanalyses were generally consistent in their simulation of the seasonal cycle and
interannual variations in regional mean ozone, with high ozone in GEOS-Chem and low ozone in CAMSRA at the surface.
In the free troposphere the reanalyses are in better agreement in their simulation of the ozone variability as well as the ozone
mixing ratio, with a few exceptions. At 500 hPa the seasonal maximum in ozone over the US occurs three months later in
GEOS-Chem than in the other reanalyses. At 250 hPa the amplitude of the seasonal cycle in ozone is larger over South
America and southern Africa in GEOS-Chem than in CAMSRA or TCR-2. At both 500 hPa and 250 hPa there are
significant differences across the reanalyses in the interannual variability in ozone over northern Africa.

Evaluation of the reanalyses with TOAR surface ozone observations reveal that GEOS-Chem, TCR-2, and CMAQ are
biased high, with surface ozone biases exceeding 10 nmol mol$^{-1}$ in GEOS-Chem. The CAMSRA product has the smallest
bias with negative biases in central, eastern, and southern Europe, and in the central and western US, and positive biases
everywhere else. We did not evaluate CAQRA with the surface ozone observations because of the limited number of
observations available in Asia in the TOAR-I database. In the free troposphere the biases in the reanalyses relative to the
ozonesonde data are small, for example, less than 4 nmol mol$^{-1}$ everywhere at 500 hPa. The correlations between the
reanalyses and the ozonesonde data were about 0.9 and 0.8 for the northern and southern midlatitudes, respectively, with
lower correlations of between 0.4–0.7 in the tropics. The higher correlations in the midlatitudes likely reflect that fact that
ozone transport in the extratropical free troposphere is controlled by large-scale synoptic processes that are well represented
by meteorological reanalyses. In contrast, in the tropics, despite the constraints that satellite observations provide on
atmospheric composition, transport of trace gases is dominated by convective processes, which are less well captured in
meteorological reanalyses.

Our results suggest that chemical reanalyses should provide valuable information for quantifying regional and
interannual variations in ozone in the free troposphere. Large regional discrepancies in ozone at the surface between the
reanalyses impact their utility for surface ozone studies. The discrepancies reflect differences in the configuration of the
assimilation schemes employed in the reanalyses as well as discrepancies in the chemical mechanisms in the models.
Improvements in the reanalyses are needed to mitigate these discrepancies by better exploiting the assimilated observations,
both satellite and in situ observations, to jointly optimize the atmospheric concentrations of ozone and its precursors together
with the ozone precursor emissions. Incorporating newly available satellite observations, such as isoprene observations from



CrIS and high temporal resolution GEO observations of ozone, NO₂, and HCHO from GEMS and TEMPO, will provide greater constraints on tropospheric ozone. This will enhance the consistency and quality of the surface ozone analyses and

thus the utility of the reanalyses for quantifying regional and long-term variations in surface ozone.

*Author contributions*. D.J., K.M., and H.W. designed the research. L.P., Z.Q., and W.C. performed the analysis. Z.Q., K.M.,

A.I., R.K., and X. T. provided chemical reanalysis products. All authors contributed to the writing of the manuscript.

*Competing interests*. The authors declare that they have no conflict of interest.


*Data Availability*. The CAMSRA data are available from https://atmosphere.copernicus.eu/data. TCR-2 data can be publicly accessed                                    at                                    https://disc.gsfc.nasa.gov/information/data-release?title=Release%20of%20TROPESS%20Chemical%20Reanalysis%20Products. The CMAQ reanalysis is available at: https://gdex.ucar.edu/dataset/382_kumar.html. The GEOS-Chem-adjoint top-down NOx emission data are available at

https://dataverse.harvard.edu/dataset.xhtml?persistentId=doi:10.7910/DVN/HVT1FO.

*Acknowledgments*. This work was supported by the Natural Science and Engineering Research Council of Canada (NSERC). Part of this work was conducted at the Jet Propulsion Laboratory, California Institute of Technology, under contract with the

NASA. The NSF National Center for Atmospheric Research is sponsored by the U.S. National Science Foundation. The TCR-2 product was generated by the calculations using the Earth Simulator with the support of the Japan Agency for Marine-Earth Science and Technology. The Copernicus Atmosphere Monitoring Service is operated by the European Centre for Medium-Range Weather Forecasts (ECMWF) on behalf of the European Commission as part of the Copernicus Programme (http://copernicus.eu).

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



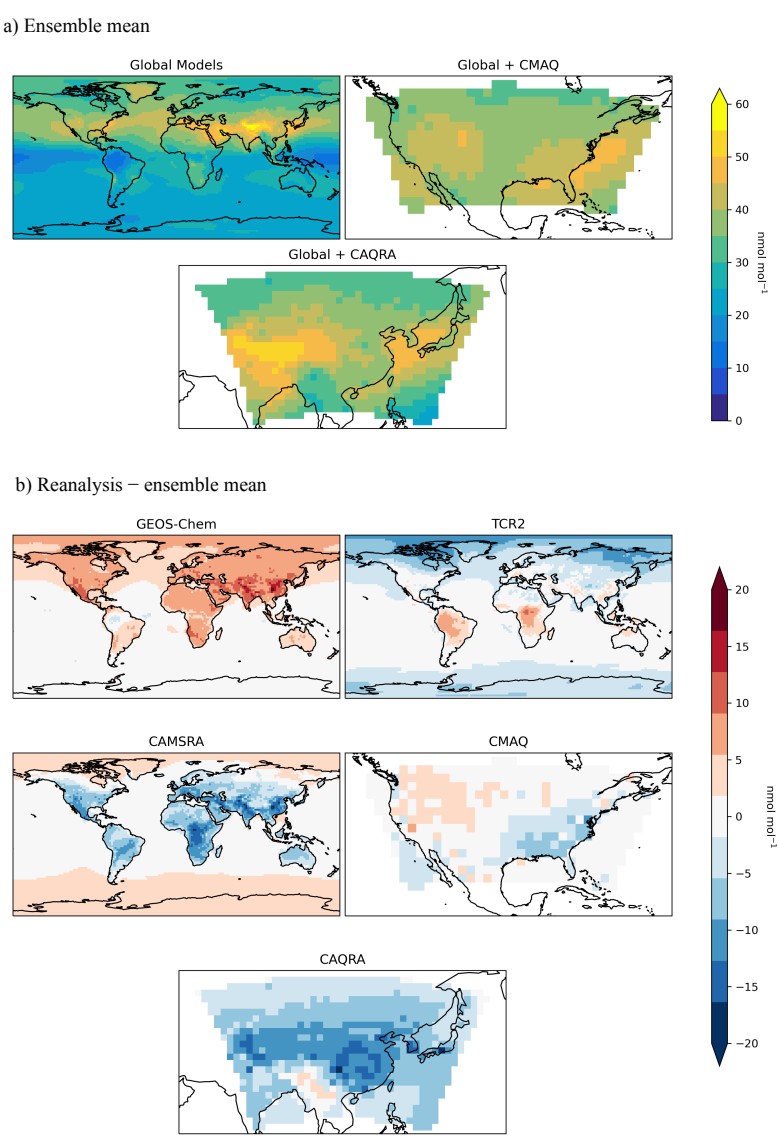

**Figure 1.** a) Ensemble mean ozone (nmol mol⁻¹) at the surface for the global reanalyses (CAMSRA, GEOS-Chem, and TCR-2) for 2006–2016 (top), the global reanalyses and CMAQ over the United States (top right), and the global reanalyses and CAQRA over Asia (bottom). b) Mean differences (nmol mol⁻¹) between surface ozone in the individual reanalyses and the ensemble mean. The ensemble mean and the differences from the mean for the CAQRA evaluation were calculated for 2013–2016 since CAQRA data are only available starting in 2013.

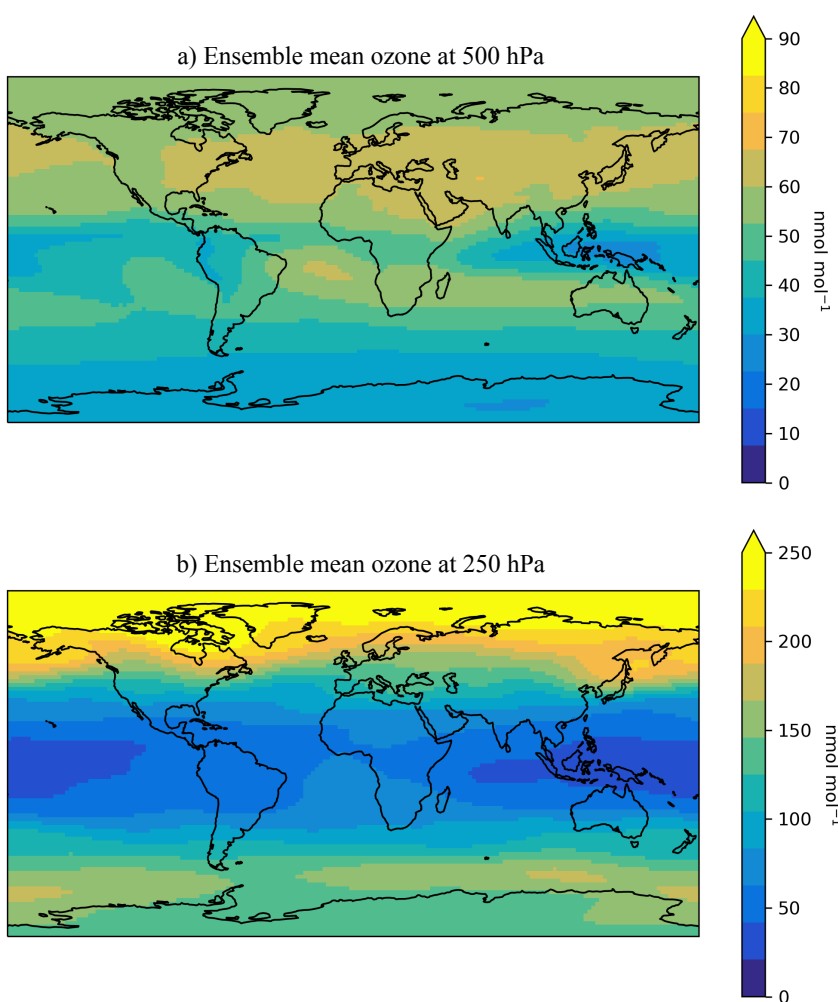

**Figure 2.** Ensemble mean ozone (nmol mol[-1]) at 500 hPa (a) and 250 hPa (b) for the three global reanalyses for 2006-2016.



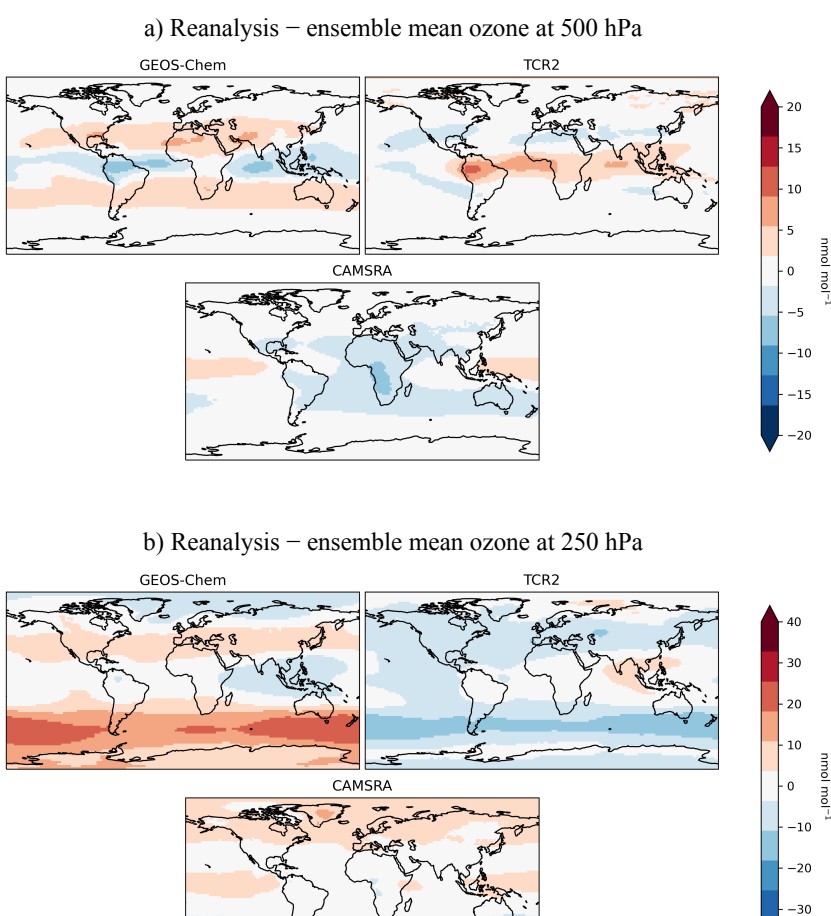

**Figure 3.** Mean differences (nmol mol⁻¹) between the individual reanalyses and the ensemble mean (shown in Figure 2) at 500 hPa (a) and 250 hPa (b).




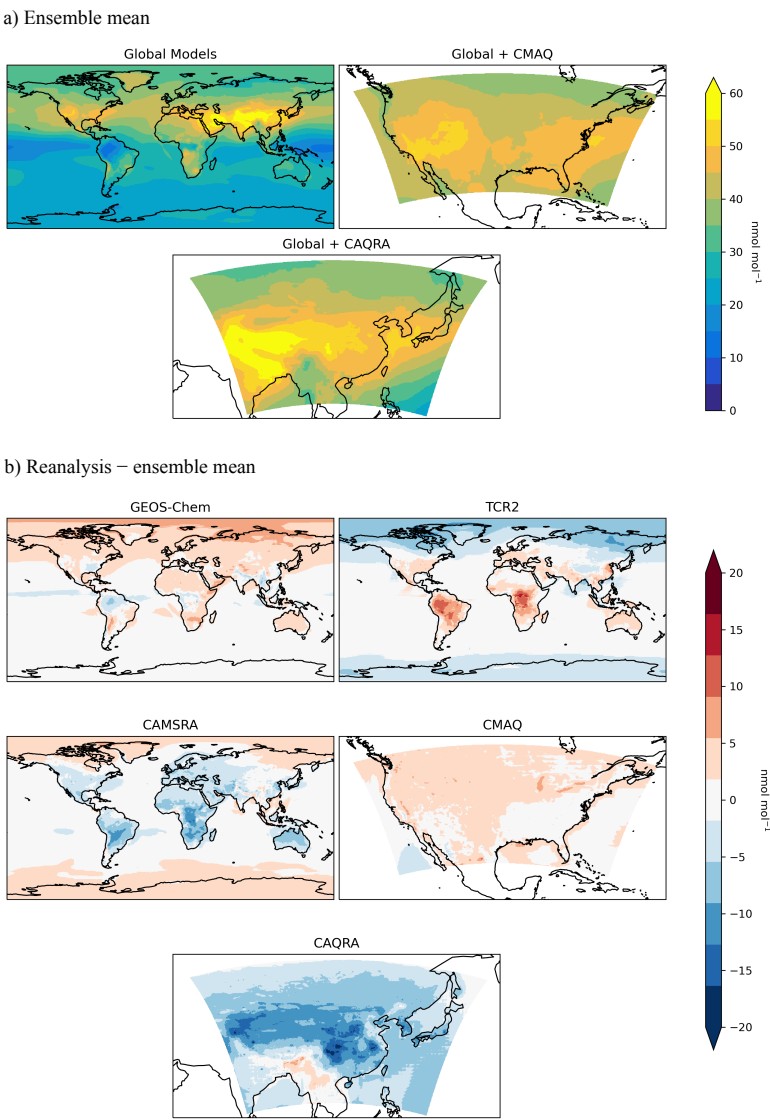

**Figure 4.** a) Ensemble mean MDA8 ozone (nmol mol⁻¹) at the surface for the global reanalyses (CAMSRA, GEOS-Chem, and TCR-2) for 2006–2016 (top), the global reanalyses and CMAQ over the United States (top right), and the global reanalyses and CAQRA over Asia (bottom). b) Mean differences (nmol mol⁻¹) between MDA8 ozone in the individual reanalyses and the ensemble mean. The ensemble mean and the differences from the mean for the CAQRA evaluation were calculated for 2013–2016 since CAQRA data are only available starting in 2013.






**Figure 5a.** Seasonal variations in regional mean ozone (nmol mol$^{-1}$) at the surface for the regions defined in Table 1. Shown are the monthly mean fields for CAMSRA (blue), GEOS-Chem (red), TCR-2 (green), CMAQ (black), and CAQRA (yellow). The monthly fields were averaged for 2006–2016, except for CAQRA, which was averaged for 2013–2016. The regional and temporal standard deviation for each month is indicated by the shading.








**Figure 5b.** As in Figure 5a, but for 500 hPa.







**Figure 5c.** As in Figure 5a, but for 250 hPa.







**Figure 6a.** Time series of regional mean ozone (nmol mol⁻¹) at the surface for 2006–2016 for the regions defined in Table 1 (and shown in Fig S4). Shown are the monthly mean fields for CAMSRA (blue), GEOS-Chem (red), TCR-2 (green), CMAQ (black), and CAQRA (yellow). The time series for the CAQRA product extends from 2013–2016.






**Figure 6b.** As in Figure 6a, but for 500 hPa.







**Figure 6c.** As in Figure 6a, but for 250 hPa.





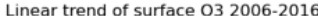

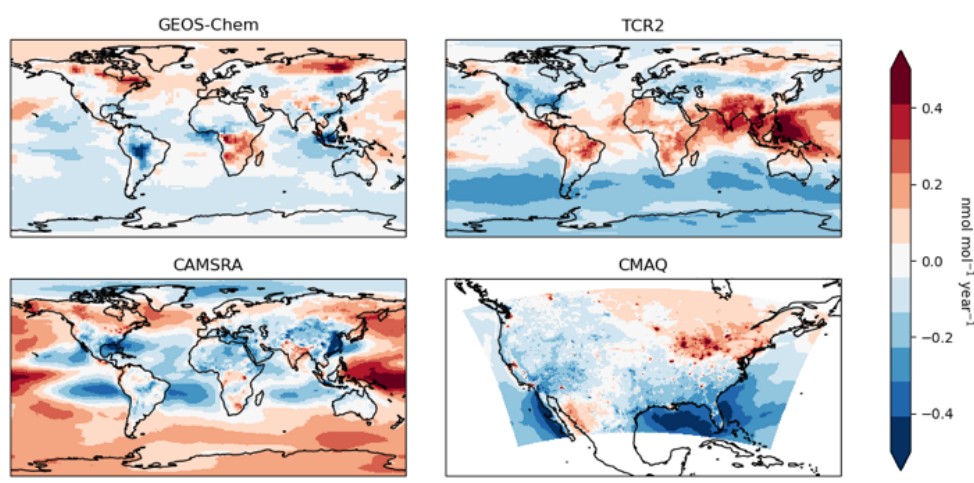

**Figure 7.** Spatial distribution of the linear trends (nmol mol⁻¹ per year) in surface ozone for 2006–2016 for the individual reanalysis. The trend was not calculated for CAQRA since the CAQRA fields are only available starting in 2013.





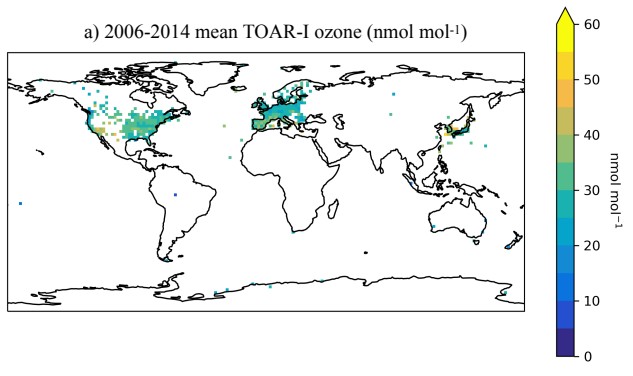

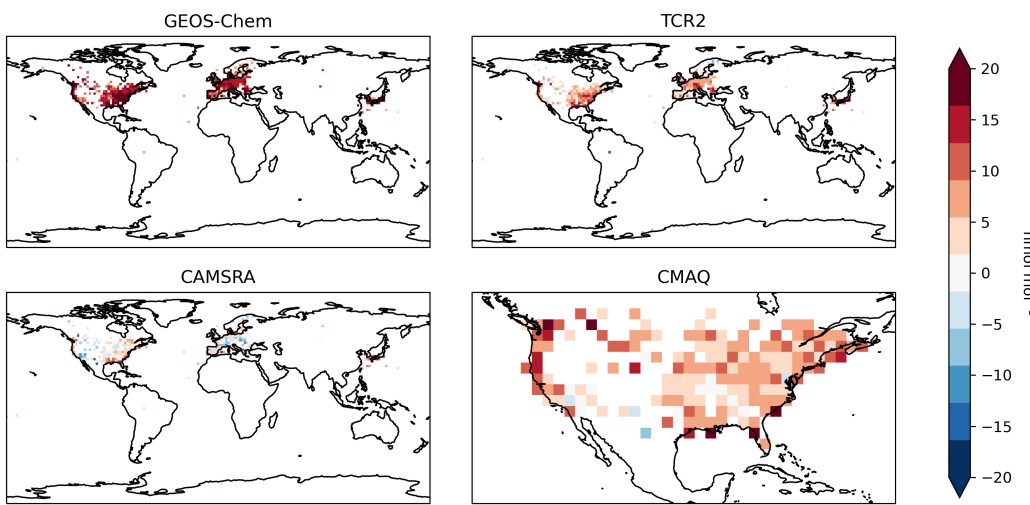

**Figure 8.** a) Mean surface ozone (nmol mol$^{-1}$) for 2006–2014 from the TOAR-I database. b) Mean bias (nmol mol$^{-1}$) between surface ozone in the individual reanalyses and the TOAR-I observations. The CAQRA product was not included in the evaluation because of the short temporal overlap between CAQRA and the TOAR-I database and the limited number of TOAR-1 observations over Asia.



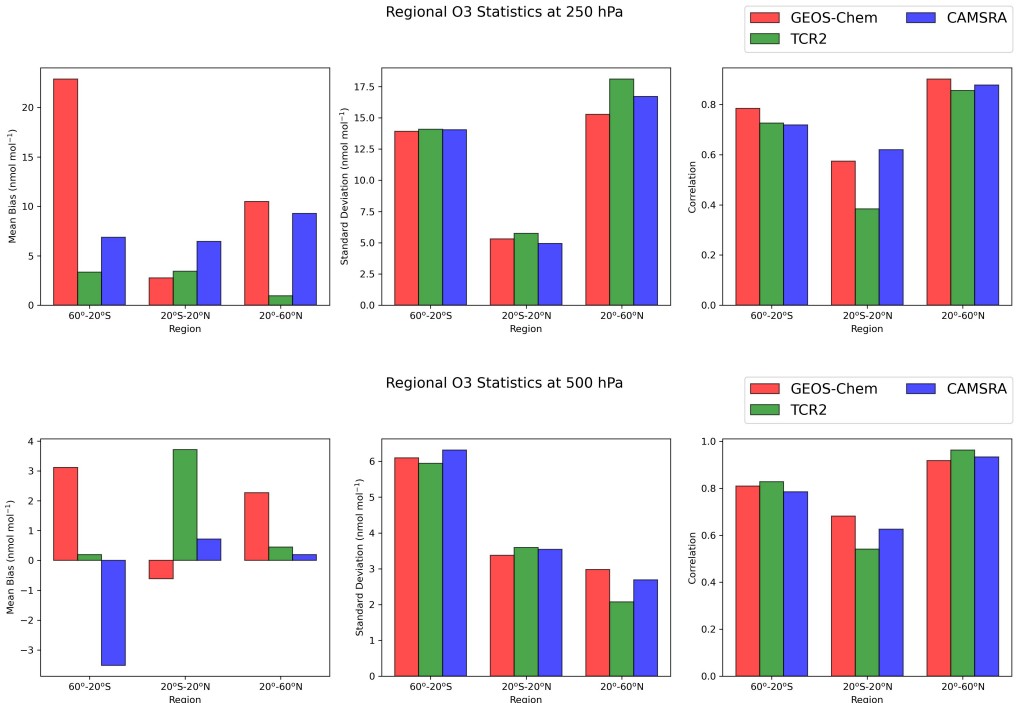

**Figure 9.** Regional statistics for the evaluation of the reanalyses with ozonesondes at 250 hPa (top row) and 500 hPa (bottom row). Shown are the mean bias (nmol mol$^{-1}$) between the reanalyses and the ozonesondes (left column), the standard deviation between the reanalyses and the ozonesondes (middle column), and the correlation between the reanalyses and the ozonesondes (right column).



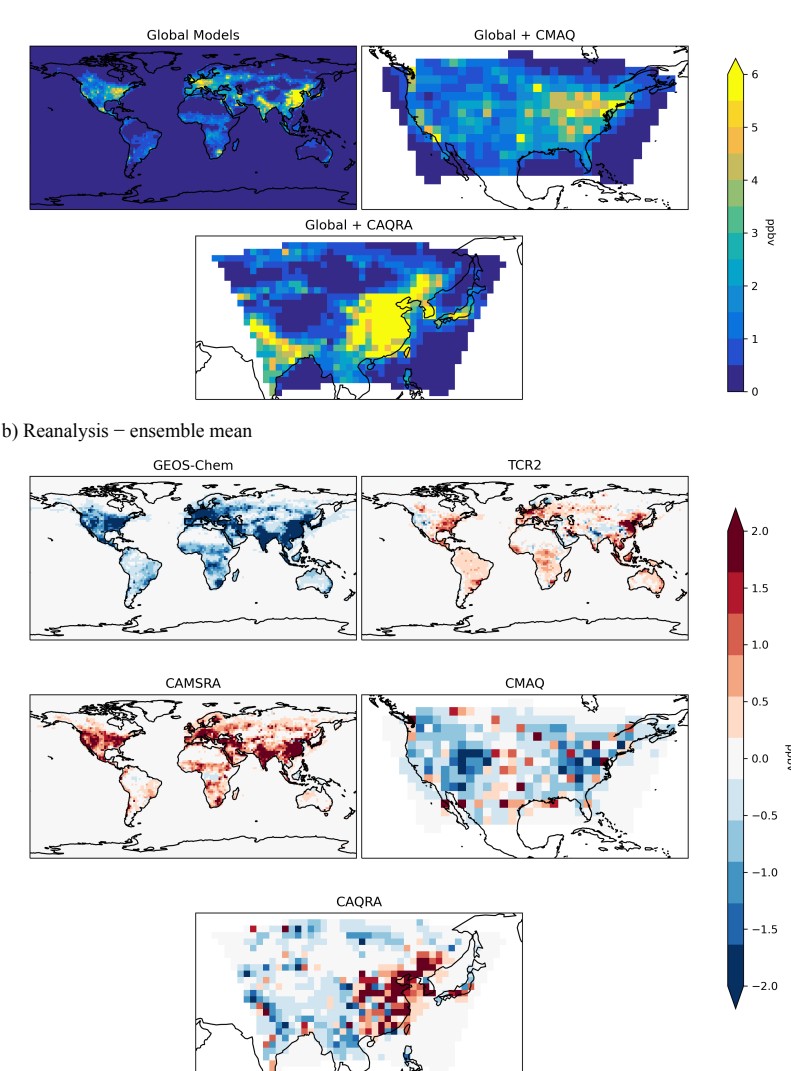

**Figure 10**. Ensemble mean NO₂ (ppbv) at the surface for the global reanalyses (CAMSRA, GEOS-Chem, and TCR-2) for 2006–2016 (top), the global reanalyses and CMAQ over the United States (top right), and the global reanalyses and CAQRA over Asia (bottom). b) Mean differences (ppbv) between surface NO₂ in the individual reanalyses and the ensemble mean. The ensemble mean and the differences from the mean for the CAQRA evaluation were calculated for 2013–2016 since CAQRA data are only available starting in 2013.





**Figure 11.** Seasonal variations in regional mean NO₂ (ppbv) at the surface for the regions defined in Table 1. Shown are the monthly mean fields for CAMSRA (blue), GEOS-Chem (red), TCR-2 (green), CMAQ (black), and CAQRA (yellow). The monthly fields were averaged for 2006–2016, except for CAQRA, which was averaged for 2013–2016. The regional and temporal standard deviation for each month is indicated by the shading.