# Peer review of "Assessment of regional and interannual variations in tropospheric ozone in chemical reanalyses"

_EGUsphere, 2024_

## Author Comment (AC1)

We thank the reviewers for their thoughtful and helpful comments on the manuscript. Below are our responses to the reviewers' comments, with each individual comment in bold italics, followed by our response.

*Reviewer 1*

*General comments:*

*Comparing and Evaluating reanalyses for atmospheric composition is an effort which, to my best knowledge, has not been done so far on such a comprehensive data basis. The authors pick up tropospheric ozone, probably the species of broadest interest for air quality and climate. While the complex interplay between atmospheric chemistry and transport, together with uncertainties in sources and sinks, makes it difficult to model this species adequately on all scales, different kind of satellite observations can be assimilated to relax the model results to observational evidence. Here, three global reanalyses and two regional reanalyses are intercompared and evaluated with independent observations from surface stations and ozonesondes. The study is generally well written, touches on a topic highly relevant for ACP and should be published with minor modifications. In the following, I suggest to take a few points into consideration:*

- *The underlying model basically differ in their chemical mechanisms, their emissions, and in their spatial resolution and extent. Some information is still missing or not well referenced in the model descriptions. It will certainly be helpful for the reader if such model basics can be summarized in a table, together with data assimilation specifics as assimilated species and satellite instruments considered.*
- *It would be intriguing to know more about the influence of the different chemical mechanisms and emission inventories. A few plots showing the most important precursor emissions (or some statistics) could shed light on the uncertainties introduced here. Ideally, one would also compare model results without data assimilation, but this will need additional model simulations which may be out of the scope of this manuscript.*
- *Unfortunately, the regional reanalyses provide only surface data. It should be feasible to include the necessary data for model evaluation also on 500 and 250 hPa for these models.*

We have revised Table 1 to include more information on the assimilated species and satellite instrument that produced the measurement.

We agree that a comparison of the model with and without data assimilation would be helpful. Unfortunately, the control run without assimilation was not available for all the reanalyses and rerunning the models for the entire analysis period was not feasible. Regarding plots of the important ozone precursors, we only have $NO_2$ archived from all of the reanalyses, which is why Section 3.4 focused on surface $NO_2$. Similarly, the full model output for the regional reanalyses were not available on the higher model levels.

*Specific comments:*

*Lines 17-20: Please state here explicitly that you are using global as well as regional reanalyses.*

We now state that these are global and regional reanalyses.

*Lines 34-41: You should spend a few more sentences on the processes regulating tropospheric ozone on different scales, particularly the complex interplay between VOC and NOx emission for ozone production, major sinks of ozone, and the role of long-range transport and Stratosphere-Troposphere exchange. Not all of these factors can be constrained by observations but need to be addressed by the chemical mechanisms and transport peculiarities of the models used. For further reading, you could refer to, e.g., Monks et al., 2015.*

We have added some additional text describing the complex interplay between NOx and VOC emissions on lines 38 to 45 in the revised manuscript. We have also added the Monks et al. (2015) reference.

*Line 39: Reference "Chang et al., 2017" is missing.*

We have added the reference.

*Line 43: Reference "EPA, 2024" is missing.*

We have added the reference.

*Lines 66-72: The influence of STE on tropospheric ozone is still under debate and variates considerably between models, as stated in, e.g., Young et al. (2018) and Griffiths et al. (2021). Please reflect here latest reviews on the topic and mention other potential sources in the upper troposphere as lightning NOx.*

Griffiths et al. (2021) examined long-term variations (between 1850-2100) in tropospheric ozone in the CMIP6 simulations, which is a different focus from the work in this manuscript. Instead, we have the added some text on lines 81-83 about the uncertainty in STE, following Young et al. (2018). We have also added some text on lines 72-73 on the source of lightning NOx.

*Line 73: "PBL" is not used furtheron in the manuscript and can be omitted.*

It has been omitted.

*Lines 79-80: Give some references here, like Inness et al. (2013).*

We have added the following references: Parrington et al., 2008; Inness et at al., 2013; Miyazaki et al., 2015; Gaubert et al., 2017; and Zhang et al., 2019.

*Lines 84-86: All five references are missing.*

Thanks for catching that. We have added the references.

*Line 91: Reference "Jiang et al., 2017" is missing. Should it be "Jiang et al., 2018"?*

It is Jiang et al. (2017), and we have added it to the reference list.

**Line 92: Reference "Cady-Pereira at al., 2017" is missing.**

Thank you. We have added the reference.

**Line 99: You could refer here to https://igacproject.org/activities/TOAR/TOAR-II .**

Done.

**Line 100 and Table 1: CONUS is first mentioned here and needs to be explained (instead of line 173).**

We have moved the definition of CONUS to this paragraph.

**Line 158: You could refer to https://gmao.gsfc.nasa.gov/reanalysis/MERRA-2/ for MERRA-2.**

We have added the Gelaro et al. (2017) reference for MERRA-2.

**Line 159: Please give numbers for species and reactions of the chemical mechanism.**

The GEOS-Chem adjoint version v35j has 43 species and 297 chemical reactions. This information has been added to the text.

**Lines 163-169: Interestingly, GEOS-Chem assimilates NO2 only. Can you give a few details on the decision not to assimilate further species?**

The GEOS-Chem reanalysis assimilated only $NO_2$ as the focus of Qu et al. (2020) was on the impact of discrepancies in OMI $NO_2$ retrievals on inferred NOx emissions and posterior ozone simulations. We now state this in the manuscript.

**Line 178: Please give numbers for species and reactions of the chemical mechanism.**

The mechanism has 147 species and 343 reactions. This information has been added to the text.

**Line 179: You could refer to https://www.epa.gov/air-emissions-inventories/2014-national-emissions-inventory-nei-data .**

We have added the URL.

**Line 180: Citation for CMAQ-SMOKE: de Almeida Albuquerque et al., (2018).**

Thank you. We have added the citation.

**Line 183: Please give a reference for BEIS.**

We have added the Vukovich and Pierce (2002) reference.

**Lines 184-190: Again, it will be interesting to get information on the choice of assimilation species.**

The CO chemical data assimilation capability in Kumar et al. (2024) was a recent addition to the previously existing MODIS AOD data assimilation capability. We now state this in the manuscript.

*Line 203: I don't understand the term "inflation factor" in the context of data assimilation. Please explain shortly its relevance or omit.*

We have decided to remove the statement since it is a detail of the assimilation configuration that is discussed in the given reference.

*Line 230: Replace "northern India" by "Tibetan plateau".*

It has been changed.

*Line 243: The NH maximum extends over large areas of subtropics and mid latitudes.*

We have changed the text to "northern subtopics and mid-latitudes".

*Line 244: The fact that only surface fields are available from the regional reanalyses is important for the soundness of the study and should be stated much earlier. What is the reason for not providing at least two additional levels at 500 and 250 hPa from these data sets?*

The full 3D fields were not archived for the regional reanalyses. We now state that "the regional reanalyses are evaluated only at the surface" in the introduction, on lines 116-117.

*Lines 246-247: I see differences of more than 5 nmol mol-1 for both reanalyses, GEOS-Chem and TCR2. Please revise. Replace "mean differences" in line 247 by "major differences"?*

The differences are generally less than 5 nmol/mol except for some localized regions in the tropics.

We have changed "mean differences" to "major differences".

The revised the text now reads:
"The major differences with respect to the ensemble mean are mainly in the tropics and subtropics, with TCR-2 and GEOS-Chem exhibiting an opposite pattern of differences. The differences in the individual reanalyses are generally less than 5 nmol mol$^{-1}$, except for localized regions in the tropics where the differences are larger."

*Lines 278-281: Be more careful in your description of Fig. 5. TCR2 is higher over South America than GEOS-Chem in all months except July. CMAQ is higher over the US than GEOS-Chem from November to May. Also, TCR2 is higher than GEOS-Chem for parts of the year.*

Thank you for catching that. Figure 5 had an error in the interpolation of the GEOS-Chem fields on the pressure levels. We have corrected the figure and revised its description in the text.

*Lines 283-286: I see discrepancies in all tropical regions. Please revise description of Fig. 5.*

Yes, we agree. In the original text we stated "In the tropics, there are larger discrepancies between the reanalyses over South America, North Africa, and the Middle East. We have revised

this to state "There are larger discrepancies across the tropics, particularly over South America, North Africa, and the Middle East."

**Lines 289-290: Same holds true for China.**

In the corrected Figure 5, the GEOS-Chem fields are more consistent with the other reanalyses in the upper troposphere over China.

**Lines 306-308: I can't see from Fig. 6 that the variability of GEOS-Chem is much different from that of the other reanalyses. There are some shifts between the curves and also differences in amplitude and variability from month to month, but without any preference for one model. How can you say that GEOS-Chem fails to reproduce year-to-year variability without knowing the ground truth?**

We agree with the reviewer. In the absence of a ground truth, it is not justified to say that GEOS-Chem fails to reproduce the year-to-year variability. We have removed this text from the manuscript.

**Lines 309-316: Are the reported trends considered to be significant in a statistical sense? If yes, please mark the respective areas in Fig. 7. If not, please omit the whole paragraph including Fig. 7.**

We have revised the figure to include stippling to indicate where the trends are statistically significant.

**Line 325: Can you add a Figure or Table providing information on the surface stations, their location and observation statistics?**

We did not produce the grided TOAR-1 surface ozone dataset and do not have the information on the observation statistics used in constructing the dataset. These data have been widely used for model evaluation since the TOAR-1 assessment. We refer the reader to the Schultz et al. (2017) paper cited in the manuscript for the details of the TOAR-1 dataset.

**Line 326: I would not use "global bias" here, as the surface observation do not have global coverage. Rather use "average bias" here.**

We have changed "global mean bias" to just "mean bias".

**Line 335: How many ozonesondes are used for each latitude band? Are all seasons well covered everywhere?**

There were 3 ozonesonde sites used in the 60°S–20°S latitude band, 11 in the 20°S–20°N band, and 17 in the 20°N–60°N band. We have added this information to the text in Section 3.3.

**Lines 338-339: There is no further discussion of single ozonesonde stations. The sentence as well as Figures S6 and S7 could be omitted.**

We have removed the sentence and Figures S6 and S7.

**Line 384: Reference "Sekiya et al., 2021"" is missing.**

Thank you for catching that. We have added the references.

***Lines 406-408: I guess all that what is currently possible to produce background error covariances is already being done. How do you envisage to produce better background statistics without having more or better observational data?***

Actually, the improvement of treatment error covariances is an active area of research in the data assimilation community. Furthermore, the use of different data assimilation schemes places different challenges on the construction of the covariances. It is beyond the scope of the manuscript to go into a discussion on the construction of background error covariances here.

***Line 418: Can you give a reference for TEMPO?***

We have added the Zoogman et al. (2017) reference.

***Line 420: Both references are missing.***

Thank you. We have added the references.

***Lines 440-441: Can you give an explanation for the shift in maximum ozone over the US between the models?***

We do not have the information necessary to identify the source of the shift in the ozone maximum. Please see our response to Reviewer 2.

***Line 459: Don't forget to mention the different emission inventories here.***

Thank you. We have changed the text to state "The discrepancies reflect differences in the configuration of the assimilation schemes employed in the reanalyses as well as discrepancies in the chemical mechanisms and prior emission inventories in the models."

***Line 475: Please point here to the exact data location, which is: https://ads.atmosphere.copernicus.eu/datasets/cams-global-reanalysis-eac4***

We have changed the URL as suggested.

***Line 651: Reference "Miyazaki et al., 2021" is not used in the manuscript and should be omitted.***

It has been deleted.

***Figures 5, 6, 11: Use of "mixing ratio" will be more precise than "concentration".***

We have changed the labels to "Mixing Ratio".

***Fig. 8 / Fig. S5: Please move the information on the regridding from Fig. S6 to Fig. 8.***

We kept the information in the supplement and added it to the Fig. 8 caption.

***Technical comments:***

*Line 54: ")" is missing after references.*

Thank you for catching that. It has been added.

*Line 175: Replace "reanalyses" by "reanalysis".*

It has been corrected.

*Line 176: Replace "… is based the …" by "… is based on the …".*

It has been corrected.

*Line 238: I guess you mean Fig. 1b here instead of Fig. 2.*

Yes, we mean Figure 1b. Thank you for catching that. It has been corrected.

*Line 266: Replace "Flemming et al., 2018" by "Fleming et al., 2018".*

Thank you. It has been corrected.

*Lines 322-323: Replace "… observations TOAR-1 …" by "… observations in the TOAR-1 …".*

It has been corrected.

*Line 401: Replace "… compared jointly …" by "… compared to jointly …".*

It has been corrected.

*Line 409: Replace "… factor influences …" by "… factor which influences …".*

It has been corrected.

*Line 477: Link is broken.*

We tested all of the links in this section and they are working. It is unclear why the link did not work for the reviewer.

*Figures 10, 11: Please use "nmol mol$^{-1}$ " as units.*

For the TOAR-2 publications it was decided to standardize the ozone units and use nmol mol$^{-1}$. However, Figures 10 and 11 are showing $NO_2$ mixing ratio and it is traditional in the community to report $NO_2$ in units of ppbv. To maintain consistency with the rest of the literature, we would prefer to keep the $NO_2$ plots in ppbv.

*General comments*

*The manuscript provides an evaluation of the potential utility of 5 chemical reanalyses (3 global, 2 regional) for quantifying regional and interannual variations in tropospheric ozone. Therefore, those reanalyses are intercompared and compared with surface ozone and free-tropospheric ozone measurements in terms of their climatological ozone distribution and regional ozone variations.*

*The manuscript certainly provides very useful information for a reader who wants to have an idea about the current state of tropospheric ozone representation by chemical reanalyses. The manuscript gives a nice overview of the differences between the chemical reanalyses and independent observations, but, unfortunately, the reasons and explanations behind those differences are often not mentioned. A lot of observations are being made, but much less interpretations are shared. When reading the manuscript, I often asked myself "why?", but in many cases, I did not find the answers in the manuscript. The authors should give more insight in those findings, or simply state that the reason is unclear or not known.*

*Re-reading the summary, you'll find that only in lines 451-455, an interpretation of the results is given, together with some general statements in lines 458-459.*

Unfortunately, we do not have all the information required to provide a clear explanation for the observed discrepancies. Obtaining a clear understanding of the source of the discrepancies is challenging given the differences in the models, differences in the assimilation schemes, and the range of observations assimilated. The focus of the manuscript was to examine the consistency of the ozone fields between the reanalyses and observations and assess the potential utility of the reanalyses for air quality studies. As we noted in the introduction, "A companion TOAR-II study by Sekiya et al. (2025) examines the impact of the choice of assimilated ozone and ozone precursor observations in the chemical reanalyses on the resulting ozone fields." The goal of the Sekiya et al. (2025) study was to complement the work presented here by conducting a series of observing system experiments to begin to answer why the reanalyses are different. However, although the Sekiya et al. (2024) study was informative, it was restricted to only 2010 and in designing the Sekiya et al. study we did not anticipate the questions that arose from this work. Our results point to the need for a follow-up study that expands on Sekiya et al. (2025) to specifically address the issues identified here.

We have added the following text on lines 117-118 in the introduction: "The objective of the work presented here is to examine the consistency of the ozone field between the reanalyses and observations to assess the potential utility of the reanalyses for air quality studies."

In the summary, we now mention the need for a follow-up study to better identify the source of the discrepancies reported in the manuscript. On lines 506-508 we state: "The reported discrepancies suggest the need for a detailed follow-up intercomparison study to better determine the influence of the prior emissions, model transport and chemistry, and choice of assimilated observations on the ozone reanalyses."

Furthermore, we have expanded the interpretation of our results where possible by citing published studies that have previously examined discrepancies in the reanalyses. In particular, we try to more closely link the discrepancies reported here with the results presented in Sekiya et al. (2025).

***Specific comments***

- ***Line 54: add ")" after Cuesta et al., 2013***

  It has been corrected.

- ***Lines 58-74: very nice introduction of the current state of atmospheric chemistry models in terms of representing the tropospheric ozone field. I would expect that the current manuscript provides more insight on the contribution of the different listed "model issues" that explain tropospheric ozone discrepancies between models and observations. A part of this introduction has been retaken in the Discussion section, but rather in (too) general terms. Were you not able to shed more light on the "model issues" responsible for discrepancies between models and with observations with the current study?***

  We are not able to conduct any sensitivity analyses to shed more light on the cause of the discrepancies. Please see our response to the general comments above.

- ***Table 1: I would include two additional columns here: one describing the emission databases, and one mentioning all the assimilated species (+satellites?) in the reanalysis system. It should also be clearly mentioned in this table that the latter two only provide surface ozone output, and not tropospheric ozone.***

  Table 1 now includes an additional column with the assimilated species and the satellite instrument that is the source of the observations.

- ***Line 221: regridded instead of regrided.***

  Thanks. It has been corrected.

- ***Lines 230-241 (description of Fig. 1b): WHY? And also, why are the GEOS-Chem biases w.r.t. the ensemble mean higher over the continents than over the oceans? What explains the CAMSRA high bias band over the high latitudes w.r.t. the ensemble mean?***

  The high ozone in GEOS-Chem could be due to the lack of assimilated ozone observations in the reanalysis. We now include the follow explanation based on the Sekiya et al. (2025) study: "Sekiya et al. (2025) found that ozone in the lower troposphere was high in the GEOS-Chem control (the model without assimilation) relative to the TCR-2 and CAMSRA controls, and that data assimilation increased ozone in CAMSRA and TCR-2, but had a minimal impact on ozone in GEOS-Chem."

- ***Lines 243-244: This "note" should have already been mentioned in Section 2!***

  This is now mentioned in Section 1.

- *Line 246: "the reanalyses are all closer to the mean in the middle troposphere". Closer compared to what? Compared to the situation at the surface (Fig. 1a)? Wouldn't it be recommended to use relative comparisons (percentages of mean biases) instead?*

Closer to the mean compared to the situation at the surface. We now clarify this in the text.

As regards the relative differences, please see our response below.

- *A comment for the entire manuscript: have the authors not considered to also report relative biases (percentage differences), complementary to the absolute biases, in their comparisons between models and with observations? Why not?*

We have added four supplementary figures with the relative biases (Figures S1, S3, S5, and S8 in the revised manuscript). In describing Figs 1, 3, 8, and 9 we now report the differences in mixing ratio as well as in percentages.

- *Lines 248-250: what is reason why TCR-2 and GEOS-Chem are exhibiting an opposite pattern of differences w.r.t. the ensemble mean in the middle tropospheric mean ozone distribution?*

We now cite the work of Miyazaki et al. (2020a) in trying to explain these differences. We have added the following text to the discussion on the differences in the middle troposphere in the revised manuscript: "The spatial distribution of the differences in GEOS-Chem and TCR-2 in the tropics and subtropic is similar to the spatial pattern of the multi-model spread in ozone in the middle troposphere reported by Miyazaki et al. (2020a) in their multi-model data assimilation analysis using four different CTMs, including GEOS-Chem and MIROC-Chem. Indeed, they found that the largest multi-model spread was over northern south America, where we see large differences in TCR-2 and GEOS-Chem. Miyazaki et al. (2020a) suggested that the large multi-model spread over northern South America could be due to the background errors not accurately capturing rapid error growth associated with deep convection and biomass burning. They also suggested that discrepancies in isoprene emissions and the isoprene oxidation chemistry could be another factor contributing to the large multi-model spread in the region."

- *Lines 260-264: explanations? Here, or in the discussion…*

The high ozone in GEOS-Chem relative to the ensemble mean is likely due to the fact that GEOS-Chem did not assimilate ozone observations, as explained in our response above. In fact, the comparison with ozonesondes in Fig. 9 shows that GEOS-Chem has a large high bias relative to the sondes between 20S-60S. We believe that it is this bias that is driving the ensemble mean and the observed differences relative to the mean. We have added this explanation to the manuscript. On line 295, in the discussion of the differences in Fig 3b, we added: "The high ozone in GEOS-Chem relative to the ensemble mean is likely due to the fact that GEOS-Chem did not assimilate ozone observations." Then on line 388, in the discussion of Fig 9, we added: "This large bias in GEOS-Chem in the southern midlatitudes could explain the high ozone in GEOS-Chem relative to the ensemble mean in the upper troposphere of the southern midlatitudes shown in Fig 3b."

- *Define MDA8 here again (it has been defined earlier in line 70, but only in the context of a study cited in the introduction, so it does not harm to define it again here). And also better argue why you find it important to consider this ozone metric in the comparison as well, next to the mean surface ozone metric.*

The text now reads: "Daily maximum 8-hour average (MDA8) ozone is a daily metric widely used for air quality standards and for ozone exposure studies (e.g., Turner et al., 2015; Fleming et al., 2018; Lyu et al., 2019; Chen et al., 2024). In particular, it is used for evaluating the mortality risks associated with short-term ozone exposure (Fleming et al., 2018). We include MDA8 in our evaluation since the ozone reanalyses could be useful for ozone exposure studies where local surface air quality observations are lacking."

- *Line 275: how have those nine regions been defined? Based on which criteria?*

The regions are based those in Miyazaki et al. (2017a). We have added this to the text.

- *3.2: looking at the plots of the seasonal variation, I would start the discussion with describing if the different reanalyses show a similar shape of the seasonal cycle, rather than with the differences in the monthly ozone mixing ratios (e.g. lines 276-281), which is (at least partly) covered by the previous figures 1-4 and S1-S3. To me, the most striking feature from Fig 5 is the quite different shape in seasonal cycle at most regions of GEOS-Chem compared to the other reanalyses. Explain this.*

We agree with the reviewer. We began the discussion in the section by noting in the second sentence that "at the surface (Fig 5a), the seasonal cycle is consistent across all the reanalyses." The main differences are in the GEOS-Chem amplitude and the fact that GEOS-Chem is shifted high, whereas CAMSRA is shifted low. The high ozone in GEOS-Chem, as discussed in our response above, is likely due to the lack of assimilated ozone. The low ozone in CAMSRA is interesting because the evaluation in Section 3.3 shows that ozone in CAMSRA is in better agreement with the TOAR-1 observations in North America and Europe.

- *Lines 291-292: Finally, some explanation, but only of some specific features (Lines 289-290). I would expect more of these explanations throughout the text.*

As mentioned in our response to the general comments above, we have added more explanation where possible, relying on previously published studies using the reanalyses.

- *Lines 293-301: The discussion of the time series of the regional mean ozone concentrations in Fig. 6 again starts with comparing the mean ozone concentrations, whereas I found especially the differences in the amplitudes of the seasonal cycle, for especially the surface ozone levels, most striking. Maybe you can first highlight this and try to come up with an explanation for this feature.*

The differences in the amplitude of the ozone seasonal cycle and the magnitude of the offsets are particularly striking for the surface. As we noted in the discussion section, the ozone lifetime is shorter at the surface than in the middle and upper troposphere, thus, the surface ozone analysis will be strongly influenced by discrepancies in the model chemistry and precursor emissions at the surface. This is consistent with the fact that the discrepancies are

smaller in the middle and upper troposphere. We now explicitly link this section of the discussion back to Fig 6. The text in the discussion now reads: "Near the surface the ozone lifetime is shorter than in the middle and upper troposphere, thus, near-surface information from the observations ingested in the assimilation will be rapidly destroyed and the surface ozone analysis will be strongly influenced by discrepancies in the model chemistry and precursor emissions at the surface. As a result, the configuration of the assimilation will have a significant impact on the surface ozone analysis. This could explain the larger discrepancies that we see between the reanalyses at the surface as compared to the middle and upper troposphere in Figs. 5 and 6."

- *Lines 301-308: Why is the worst agreement found over Northern Africa (Fig. 6b and 6c)? Why does TCR-2 significantly overestimate the ozone minima? Explain why the simulated variability is most different in GEOS-Chem.*

  It is important to note that the Northern African region is actually tropical northern Africa. Furthermore, as can be seen in Figs. 6b there are also large discrepancies over South America and Central Africa. These are all regions where convective transport is dominant. Given the high variability associated with convection, we would expect it to be much more challenging for the reanalyses to constrain ozone in these regions. These regions are also strongly influenced by biomass burning emissions and biogenic ozone precursor emissions, which are poorly constrained in models. We have added this discussion about the influence of convection to the manuscript on line 343.

- *Lines 309-316: the authors might add which (sign of) trends are expected, based on observations, in the different regions, rather than summing up the trend differences calculated from the different reanalyses for the different regions.*

  We focused on the consistency of the trends across the reanalyses. We are reluctant to qualitatively compare the trends with reported trends based on observations without first sampling the reanalyses at the times and locations of the observations on which the reported trends are based, since even the sign of the trend can change based on the observational sampling.

- *Section 3.3: should the discussion in this section not benefit from including the relative (percentage) mean differences, especially if you compare the mean biases between different seasons (e.g. lines 330-334), and at different latitudes (e.g. lines 342-351).*

  We now include the biases in percentage.

- *Fig. 9: perhaps add the upper and lower bounds of the found correlation coefficients for the different latitude bands and reanalysis, so that the reader gets an idea about their variability.*

  We do not calculate the correlations for individual ozonesonde sites, so we cannot report upper and lower bounds on the correlations in each region. Instead, the reanalyses and the observations were aggregated for each latitude band to construct a time series for each reanalysis and the observations in each region. The correlation coefficients were calculated from those time series.

- *Section 3.4: it is a very nice add-on that you also look at the surface NO2 distribution, but rather it should be better explained what the figures 10 and 11 can learn us (or not) to interpret the surface ozone differences between the reanalyses, as shown in figures 1 and 5.*

Our discussion in this section was focused on the possible factors that can contribute to differences in surface $NO_2$, and thus ozone in the reanalyses. We have expanded the discussion in this section by citing the observing system experiments by Sekiya et al. (2025). We have added the following text to the end of Section 3.4 (on line 410): "In their observing system experiments, Sekiya et al. (2025) found that assimilation of the satellite NO2 data produced a larger increase in lower tropospheric ozone in TCR-2 than in GEOS-Chem, which they suggested could be due to differences in the sensitivities of ozone to changes in NOx emissions in GEOS-Chem and TCR-2, as reported by Miyazaki et al. (2020a) in the multi-model data assimilation system. Sekiya et al. (2025) also noted that Qu et al. (2020) reported lower posterior NOx emission estimates in GEOS-Chem when assimilating the NASA standard OMNO2 $NO_2$ product compared to when assimilating the KNMI DOMINO product (Boersma et al., 2011) or the QA4ECV product (Boersma et al., 2018). The GEOS-Chem results presented here are based on the OMNO2 product."

- *Section 4 (Discussion): very general explanations (model resolution, different chemistry, assimilation configuration) are given, often citing from literature, to explain (surface) ozone differences between the different reanalyses. A direct linking to many of the in great detail reported discrepancies between the reanalyses earlier in the paper is missing (there and in this discussion section). This is somewhat disappointing for the reader. It might be that the authors do not have the information or knowledge to dig deeper into those discrepancies, because many contributing factors intervene here. If so, please state this and be clearer on that.*

As discussed in our response to the general comments above, we do not have the information to dig deeper into these discrepancies. As we now mention in the summary, our results suggest the need for a detailed follow-up intercomparison study to identify the factors contributing to these differences.

- *Section 5 (Summary): as already mentioned at the end of my general comments, there is very little interpretation included in the summary.*

Please see our response to the general comments above.

- *Data availability section: please, also mention where the TOAR-I surface ozone data and the used ozonesonde can be obtained.*

We have added links to the sites for the ozonesonde and TOAR surface ozone data.

- *Supplementary Figures S6: What time series are represented here? Monthly means? Or have the reanalyses been sampled to the ozonesonde observations? Please specify! How have the ozonesonde graphs been ordered? Alphabetically? Why not e.g. with decreasing latitude? It also seems that there are some outliers in the ozonesonde observations (e.g. at*

*Fiji, Lauder, Paramaribo, Resolute, Samao, Scoresbysund, Sodankyla, South Pole) that possibly might be removed to make the comparisons clearer.*

We have removed Fig 6, as suggested by Reviewer 1.